# Generating Adjacency-Constrained Subgoals in Hierarchical Reinforcement Learning

**Tianren Zhang**[*,1]**, Shangqi Guo**[*,1]**, Tian Tan**[2]**, Xiaolin Hu**[†,3,4,5]**, Feng Chen**[†,1,6,7]

[1] Department of Automation, Tsinghua University
[2] Department of Civil and Environmental Engineering, Stanford University
[3] Department of Computer Science and Technology, Tsinghua University
[4] Beijing National Research Center for Information Science and Technology
[5] State Key Laboratory of Intelligent Technology and Systems
[6] Beijing Innovation Center for Future Chip
[7] LSBDPA Beijing Key Laboratory
{zhang-tr19,gsq15}@mails.tsinghua.edu.cn; tiantan@stanford.edu;
{xlhu,chenfeng}@mail.tsinghua.edu.cn

## Abstract

Goal-conditioned hierarchical reinforcement learning (HRL) is a promising approach for scaling up reinforcement learning (RL) techniques. However, it often suffers from training inefficiency as the action space of the high-level, i.e., the goal space, is often large. Searching in a large goal space poses difficulties for both high-level subgoal generation and low-level policy learning. In this paper, we show that this problem can be effectively alleviated by restricting the high-level action space from the whole goal space to a $k$-step adjacent region of the current state using an adjacency constraint. We theoretically prove that the proposed adjacency constraint preserves the optimal hierarchical policy in deterministic MDPs, and show that this constraint can be practically implemented by training an adjacency network that can discriminate between adjacent and non-adjacent subgoals. Experimental results on discrete and continuous control tasks show that incorporating the adjacency constraint improves the performance of state-of-the-art HRL approaches in both deterministic and stochastic environments.[1]

## 1   Introduction

Hierarchical reinforcement learning (HRL) has shown great potentials in scaling up reinforcement learning (RL) methods to tackle large, temporally extended problems with long-term credit assignment and sparse rewards [39, 31, 2]. As one of the prevailing HRL paradigms, goal-conditioned HRL framework [5, 37, 20, 42, 26, 22], which comprises a high-level policy that breaks the original task into a series of subgoals and a low-level policy that aims to reach those subgoals, has recently achieved significant success. However, the effectiveness of goal-conditioned HRL relies on the acquisition of effective and semantically meaningful subgoals, which still remains a key challenge.

As the subgoals can be interpreted as high-level actions, it is feasible to directly train the high-level policy to generate subgoals using external rewards as supervision, which has been widely adopted in previous research [26, 25, 22, 20, 42]. Although these methods require little task-specific design, they often suffer from training inefficiency. This is because the action space of the high-level, i.e., the

---

[*]Equal contribution.

[†]Corresponding authors: Xiaolin Hu and Feng Chen.

[1]Code is available at `https://github.com/trzhang0116/HRAC`.

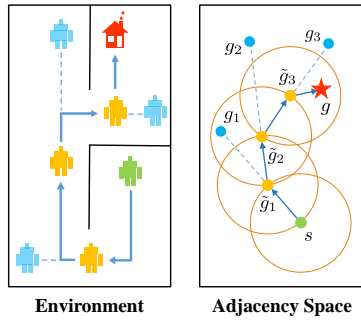

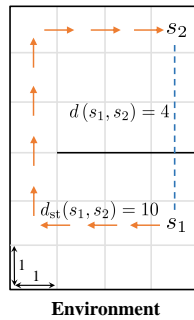

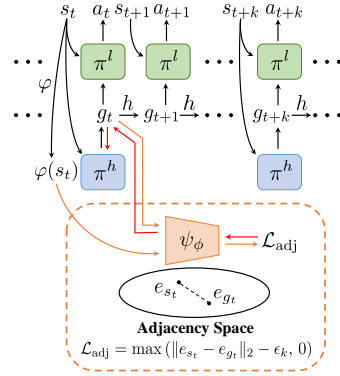

Figure 1: High-level illustration of our method: distant subgoals $g_1$, $g_2$, $g_3$ (blue) can be surrogated by closer subgoals $\tilde{g}_1$, $\tilde{g}_2$, $\tilde{g}_3$ (yellow) that fall into the $k$-step adjacent regions.

Figure 2: Comparison between shortest transition distance $d_{\mathrm{st}}$ and Euclidean distance $d$ in a toy environment.

Figure 3: The goal-conditioned HRL framework and the $k$-step adjacency constraint implemented by the adjacency network $\psi_\phi$ (dashed orange box).

goal space, is often as large as the state space. The high-level exploration in such a large action space results in inefficient learning. As a consequence, the low-level training also suffers as the agent tries to reach every possible subgoal produced by the high-level policy.

One effective way for handling large action spaces is action space reduction or action elimination. However, it is difficult to perform action space reduction in general scenarios without additional information, since a restricted action set may not be expressive enough to form the optimal policy. There has been limited literature [44, 41, 19] studying action space reduction in RL, and to our knowledge, there is no prior work studying action space reduction in HRL, since the information loss in the goal space can lead to severe performance degradation [25].

In this paper, we present an optimality-preserving high-level action space reduction method for goal-conditioned HRL. Concretely, we show that the high-level action space can be restricted from the whole goal space to a $k$-step adjacent region centered at the current state. Our main intuition is depicted in Figure 1: distant subgoals can be substituted by closer subgoals, as long as they drive the low-level to move towards the same "direction". Therefore, given the current state $s$ and the subgoal generation frequency $k$, the high-level only needs to explore in a subset of subgoals covering states that the low-level can possibly reach within $k$ steps. By reducing the action space of the high-level, the learning efficiency of both the high-level and the low-level can be improved: for the high-level, a considerably smaller action space relieves the burden of exploration and value function approximation; for the low-level, adjacent subgoals provide a stronger learning signal as the agent can be intrinsically rewarded with a higher frequency for reaching these subgoals. Formally, we introduce a $k$-step adjacency constraint for high-level action space reduction, and theoretically prove that the proposed constraint preserves the optimal hierarchical policy in deterministic MDPs. Also, to practically implement the constraint, we propose to train an adjacency network so that the $k$-step adjacency between all states and subgoals can be succinctly derived.

We benchmark our method on various tasks, including discrete control and planning tasks on grid worlds and challenging continuous control tasks based on the MuJoCo simulator [40], which have been widely used in HRL literature [26, 22, 25, 11]. Experimental results exhibit the superiority of our method on both sample efficiency and asymptotic performance compared with the state-of-the-art HRL approach HIRO [26], demonstrating the effectiveness of the proposed adjacency constraint.

## 2 Preliminaries

We consider a finite-horizon, goal-conditioned Markov Decision Process (MDP) defined as a tuple $\langle \mathcal{S}, \mathcal{G}, \mathcal{A}, \mathcal{P}, \mathcal{R}, \gamma \rangle$, where $\mathcal{S}$ is a state set, $\mathcal{G}$ is a goal set, $\mathcal{A}$ is an action set, $\mathcal{P} : \mathcal{S} \times \mathcal{A} \times \mathcal{S} \to \mathbb{R}$ is a state transition function, $\mathcal{R} : \mathcal{S} \times \mathcal{A} \to \mathbb{R}$ is a reward function, and $\gamma \in [0, 1)$ is a discount factor. Following prior work [20, 42, 26], we consider a framework comprising two hierarchies: a high-level

controller with policy $\pi^h_{\theta_h}(g|s)$ and a low-level controller with policy $\pi^l_{\theta_l}(a|s,g)$ parameterized by two function approximators, e.g. neural networks with parameters $\theta_h$ and $\theta_l$ respectively, as shown in Figure 3. The high-level controller aims to maximize the external reward and generates a high-level action, i.e. a subgoal $g_t \sim \pi^h_{\theta_h}(g|s_t) \in \mathcal{G}$ every $k$ time steps when $t \equiv 0 \,(\mathrm{mod}\ k)$, where $k > 1$ is a pre-determined hyper-parameter. It modulates the behavior of the low-level policy by intrinsically rewarding the low-level for reaching these subgoals. The low-level aims to maximize the intrinsic reward provided by the high-level, and performs a primary action $a_t \sim \pi^l_{\theta_l}(a|s_t,g_t) \in \mathcal{A}$ at every time step. Following prior methods [26, 1], we consider a goal space $\mathcal{G}$ which is a sub-space of $\mathcal{S}$ with a known mapping function $\varphi : \mathcal{S} \to \mathcal{G}$. When $t \not\equiv 0 \,(\mathrm{mod}\ k)$, a pre-defined goal transition process $g_t = h(g_{t-1}, s_{t-1}, s_t)$ is utilized. We adopt directional subgoals that represent the differences between desired states and current states [42, 26], where the goal transition function is set to $h(g_{t-1}, s_{t-1}, s_t) = g_{t-1} + s_{t-1} - s_t$. The reward function of the high-level policy is defined as:

$$r^h_{kt} = \sum_{i=kt}^{kt+k-1} \mathcal{R}(s_i, a_i), \quad t = 0,\,1,\,2,\,\cdots, \tag{1}$$

which is the accumulation of the external reward in the time interval $[kt, kt + k - 1]$.

While the high-level controller is motivated by the environmental reward, the low-level controller has no direct access to this external reward. Instead, the low-level is supervised by the intrinsic reward that describes subgoal-reaching performance, defined as $r^l_t = -D\left(g_t, \varphi(s_{t+1})\right)$, where $D$ is a binary or continuous distance function. In practice, we employ Euclidean distance as $D$.

The goal-conditioned HRL framework above enables us to train high-level and low-level policies concurrently in an end-to-end fashion. However, it often suffers from training inefficiency due to the unconstrained subgoal generation process, as we have mentioned in Section 1. In the following section, we will introduce the $k$-step adjacency constraint to mitigate this issue.

## 3 Theoretical Analysis

In this section, we provide our theoretical results and show that the optimality can be preserved when learning a high-level policy with $k$-step adjacency constraint. We begin by introducing a distance measure that can decide whether a state is "close" to another state. In this regard, common distance functions such as the Euclidean distance are not suitable, as they often cannot reveal the real structure of the MDP. Therefore, we introduce *shortest transition distance*, which equals to the minimum number of steps required to reach a target state from a start state, as shown in Figure 2. In stochastic MDPs, the number of steps required is not a fixed number, but a distribution conditioned on a specific policy. In this case, we resort to the notion of *first hit time* [43] from stochastic processes, and define the shortest transition distance by minimizing the expected first hit time over all possible policies.

**Definition 1.** *Let $s_1,\,s_2 \in \mathcal{S}$. Then, the shortest transition distance from $s_1$ to $s_2$ is defined as:*

$$d_{\mathrm{st}}(s_1, s_2) := \min_{\pi \in \Pi} \mathbb{E}[\mathcal{T}_{s_1 s_2}|\pi] = \min_{\pi \in \Pi} \sum_{t=0}^{\infty} t P(\mathcal{T}_{s_1 s_2} = t|\pi), \tag{2}$$

*where $\Pi$ is the complete policy set and $\mathcal{T}_{s_1 s_2}$ denotes the first hit time from $s_1$ to $s_2$.*

The shortest transition distance is determined by a policy that connects states $s_1$ and $s_2$ in the most efficient way, which has also been studied by several prior work [10, 8]. This policy is optimal in the sense that it requires the minimum number of steps to reach state $s_2$ from state $s_1$. Compared with the dynamical distance [15], our definition here does not rely on a specific non-optimal policy. Also, we do not assume that the environment is reversible, i.e. $d_{\mathrm{st}}(s_1, s_2) = d_{\mathrm{st}}(s_2, s_1)$ does not hold for all pairs of states. Therefore, the shortest transition distance is a quasi-metric as it does not satisfy the symmetry condition. However, this limitation does not affect the following analysis as we only need to consider the transition from the start state to the goal state without the reversed transition.

Given the definition of the shortest transition distance, we now formulate the property of an optimal (deterministic) goal-conditioned policy $\pi^* : \mathcal{S} \times \mathcal{G} \to \mathcal{A}$ [36]. We have:

$$\pi^*(s, g) \in \arg\min_{a \in \mathcal{A}} \sum_{s' \in \mathcal{S}} P(s'|s, a)\, d_{\mathrm{st}}\left(s', \varphi^{-1}(g)\right), \forall s \in \mathcal{S},\, g \in \mathcal{G}, \tag{3}$$

where $\varphi^{-1} : \mathcal{G} \to \mathcal{S}$ is the known inverse mapping of $\varphi$. We then consider the goal-conditioned HRL framework with high-level action frequency $k$. Different from a flat goal-conditioned policy, in this setting the low-level policy is required to reach the subgoals with $k$ limited steps. As a result, only a subset of the original states can be reliably reached even with an optimal goal-conditioned policy. We introduce the notion of $k$-*step adjacent region* to describe the set of subgoals mapped from this reachable subset of states.

**Definition 2.** *Let $s \in \mathcal{S}$. Then, the $k$-step adjacent region of $s$ is defined as:*

$$\mathcal{G}_A(s, k) := \{g \in \mathcal{G} \,|\, d_{\text{st}}\left(s, \varphi^{-1}(g)\right) \leq k\}. \tag{4}$$

Harnessing the property of $\pi^*$, we can show that in deterministic MDPs, given an optimal low-level policy $\pi^{l*} = \pi^*$, subgoals that fall in the $k$-step adjacent region of the current state can represent all optimal subgoals in the whole goal space in terms of the induced $k$-step low-level action sequence. We summarize this finding in the following theorem.

**Theorem 1.** *Let $s \in \mathcal{S}$, $g \in \mathcal{G}$ and let $\pi^*$ be an optimal goal-conditioned policy. Under the assumptions that the MDP is deterministic and that the MDP states are strongly connected, for all $k \in \mathbb{N}_+$ satisfying $k \leq d_{\text{st}}(s, \varphi^{-1}(g))$, there exists a surrogate goal $\tilde{g}$ such that:*

$$\begin{aligned} &\tilde{g} \in \mathcal{G}_A(s, k), \\ &\pi^*(s_i, \tilde{g}) = \pi^*(s_i, g), \; \forall s_i \in \tau \, (i \neq k), \end{aligned} \tag{5}$$

*where $\tau := (s_0, s_1, \cdots, s_k)$ is the $k$-step state trajectory starting from state $s_0 = s$ under $\pi^*$ and $g$.*

Theorem 1 suggests that the $k$-step low-level action sequence generated by an optimal low-level policy conditioned on a distant subgoal can be induced using a subgoal that is closer. Naturally, we can generalize this result to a two-level goal-conditioned HRL framework, where the low-level is actuated not by a single subgoal, but by a subgoal sequence produced by the high-level policy.

**Theorem 2.** *Given the high-level action frequency $k$ and the high-level planning horizon $T$, for $s \in \mathcal{S}$, let $\rho^* = (g_0, g_k, \cdots, g_{(T-1)k})$ be the high-level subgoal trajectory starting from state $s_0 = s$ under an optimal high-level policy $\pi^{h*}$. Also, let $\tau^* = (s_0, s_k, s_{2k}, \cdots, s_{Tk})$ be the high-level state trajectory under $\rho^*$ and an optimal low-level policy $\pi^{l*}$. Then, there exists a surrogate subgoal trajectory $\tilde{\rho}^* = (\tilde{g}_0, \tilde{g}_k, \cdots, \tilde{g}_{(T-1)k})$ such that:*

$$\begin{aligned} &\tilde{g}_{kt} \in \mathcal{G}_A(s_{kt}, k), \\ &Q^*(s_{kt}, \tilde{g}_{kt}) = Q^*(s_{kt}, g_{kt}), \quad t = 0, 1, \cdots, T - 1, \end{aligned} \tag{6}$$

*where $Q^*$ is the optimal high-level $Q$-function under policy $\pi^{h*}$.*

Theorem 1 and 2 show that we can constrain the high-level action space to state-wise $k$-step adjacent regions without the loss of optimality. We formulate the high-level objective incorporating this $k$-step adjacency constraint as:

$$\begin{aligned} &\max_{\theta_h} \quad \mathbb{E}_{\pi^h_{\theta_h}} \sum_{t=0}^{T-1} \gamma^t r^h_{kt} \\ &\text{subject to} \quad d_{\text{st}}\left(s_{kt}, \varphi^{-1}(g_{kt})\right) \leq k, \quad t = 0, 1, \cdots, T - 1 \end{aligned} \tag{7}$$

where $r^h_{kt}$ is the high-level reward defined by Equation (1) and $g_{kt} \sim \pi^h_{\theta_h}(g|s_{kt})$.

In practice, Equation (7) is hard to optimize due to the strict constraint. Therefore, we employ relaxation methods and derive the following un-constrained optimizing objective:

$$\max_{\theta_h} \quad \mathbb{E}_{\pi^h_{\theta_h}} \sum_{t=0}^{T-1} \left[ \gamma^t r^h_{kt} - \eta \cdot H\left(d_{\text{st}}\left(s_{kt}, \varphi^{-1}(g_{kt})\right), k\right) \right], \tag{8}$$

where $H(x, k) = \max(x/k - 1, 0)$ is a hinge loss function and $\eta$ is a balancing coefficient.

One limitation of our theoretical results is that the theorems are derived in the context of deterministic MDPs. However, these theorems are instructive for practical algorithm design in general cases, and the deterministic assumption has also been exploited by some prior works that investigate distance metrics in MDPs [15, 3]. Also, we note that many real-world applications can be approximated as environments with deterministic dynamics where the stochasticity is mainly induced by noise. Hence, we may infer that the adjacency constraint could preserve a near-optimal policy when the magnitude of noise is small. Empirically, we show that our method is robust to certain types of stochasticity (see Section 5 for details), and we leave rigorous theoretical analysis for future work.

# 4   HRL with Adjacency Constraint

Although we have formulated the adjacency constraint in Section 3, the exact calculation of the shortest transition distance $d_{\mathrm{st}}(s_1, s_2)$ between two arbitrary states $s_1, s_2 \in \mathcal{S}$ remains complex and non-differentiable. In this section, we introduce a simple method to collect and aggregate the adjacency information from the environment interactions. We then train an adjacency network using the aggregated adjacency information to approximate the shortest transition distance $d_{\mathrm{st}}(s_1, s_2)$ in a parameterized form, which enables a practical optimization of Equation (8).

## 4.1   Parameterized Approximation of Shortest Transition Distances

As shown in prior research [30, 10, 8, 15], accurately computing the shortest transition distance is not easy and often has the same complexity as learning an optimal low-level goal-conditioned policy. However, from the perspective of goal-conditioned HRL, we do not need a perfect shortest transition distance measure or a low-level policy that can reach any distant subgoals. Instead, only a discriminator of $k$-step adjacency is needed, and it is enough to learn a low-level policy that can reliably reach nearby subgoals (more accurately, subgoals that fall into the $k$-step adjacent region of the current state) rather than all potential subgoals in the goal space.

Given the above, here we introduce a simple approach to determine whether a subgoal satisfies the $k$-step adjacency constraint. We first note that Equation (2) can be approximated as follows:

$$d_{\mathrm{st}}(s_1, s_2) \approx \min_{\pi \in \{\pi_1, \pi_2, \cdots, \pi_n\}} \sum_{t=0}^{\infty} t P(\mathcal{T}_{s_1 s_2} = t | \pi), \tag{9}$$

where $\{\pi_1, \pi_2, \cdots, \pi_n\}$ is a finite policy set containing $n$ different deterministic policies. Obviously, if these policies are diverse enough, we can effectively approximate the shortest transition distance with a sufficiently large $n$. However, training a set of diverse policies separately is costly, and using one single policy to approximate the policy set ($n = 1$) [34, 35] often leads to non-optimality. To handle this difficulty, we exploit the fact that the low-level policy itself changes over time during the training procedure. We can thus build a policy set by sampling policies that emerge in different training stages. To aggregate the adjacency information gathered by multiple policies, we propose to explicitly memorize the adjacency information by constructing a binary *k-step adjacency matrix* of the explored states. The adjacency matrix has the same size as the number of explored states, and each element represents whether two states are $k$-step adjacent. In practice, we use the agent's trajectories, where the temporal distances between states can indicate their adjacency, to construct and update the adjacency matrix online. More details are in the supplementary material.

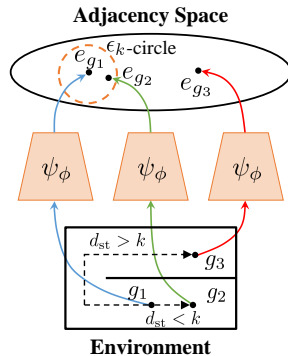

Figure 4: The functionality of the adjacency network. The $k$-step adjacent region is mapped to an $\epsilon_k$-circle in the adjacency space, where $e_{g_i} = \psi_\theta(g_i)$, $i = 1, 2, 3$.

In practice, using an adjacency matrix is not enough as this procedure is non-differentiable and cannot generalize to newly-visited states. To this end, we further distill the adjacency information stored in a constructed adjacency matrix into an adjacency network $\psi_\phi$ parameterized by $\phi$. The adjacency network learns a mapping from the goal space to an adjacency space, where the Euclidean distance between the state and the goal is consistent with their shortest transition distance:

$$\tilde{d}_{\mathrm{st}}(s_1, s_2 | \phi) := \frac{k}{\epsilon_k} \|\psi_\phi(g_1) - \psi_\phi(g_2)\|_2 \approx d_{\mathrm{st}}(s_1, s_2), \tag{10}$$

where $g_1 = \varphi(s_1)$, $g_2 = \varphi(s_2)$ and $\epsilon_k$ is a scaling factor. As we have mentioned above, it is hard to regress the Euclidean distance in the adjacency space to the shortest transition distance accurately, and we only need to ensure a binary relation for implementing the adjacency constraint, i.e., $\|\psi_\phi(g_1) - \psi_\phi(g_2)\|_2 > \epsilon_k$ for $d_{\mathrm{st}}(s_1, s_2) > k$, and $\|\psi_\phi(g_1) - \psi_\phi(g_2)\|_2 < \epsilon_k$ for $d_{\mathrm{st}}(s_1, s_2) < k$, as shown in Figure 4. Inspired by modern metric learning approaches [14], we adopt a contrastive-like loss function for this distillation process:

$$\begin{aligned}
\mathcal{L}_{\mathrm{dis}}(\phi) = \mathbb{E}_{s_i, s_j \in \mathcal{S}} \big[ & l \cdot \max\left(\|\psi_\phi(g_i) - \psi_\phi(g_j)\|_2 - \epsilon_k, 0\right) \\
& + (1 - l) \cdot \max\left(\epsilon_k + \delta - \|\psi_\phi(g_i) - \psi_\phi(g_j)\|_2, 0\right) \big],
\end{aligned} \tag{11}$$

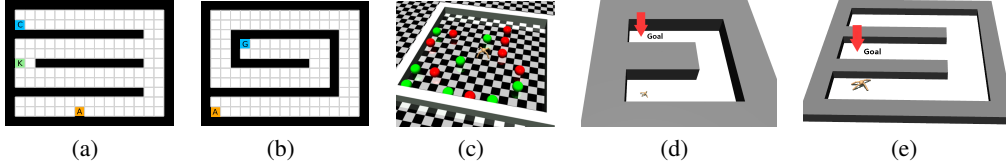

|  (a)  |  (b)  |  (c)  |  (d)  |  (e)  |

Figure 5: Environments used in our experiments. **(a)** Key-Chest. The agent (A) starts from a random position and needs to pick up the key (K) first, then uses the key to open the chest (C). **(b)** Maze. The agent (A) starts from a fixed position and needs to reach the final goal (G) with dense rewards. **(c)** Ant Gather. The ant robot starts from a fixed position and needs to collect apples (green) and avoid bombs (red) (figure is adapted from Duan et al. [6]). **(d)** Ant Maze. The ant robot starts from a fixed position and needs to reach a target position in a maze with dense rewards. **(e)** Ant Maze Sparse. The ant robot starts from a random position and needs to reach a target position in a maze with sparse rewards.

where $g_i = \varphi(s_i)$, $g_j = \varphi(s_j)$, and a hyper-parameter $\delta > 0$ is used to create a gap between the embeddings. $l \in \{0, 1\}$ represents the label indicating $k$-step adjacency derived from the $k$-step adjacency matrix. Equation (11) penalizes adjacent state embeddings ($l = 1$) with large Euclidean distances in the adjacency space and non-adjacent state embeddings ($l = 0$) with small Euclidean distances. In practice, we use states evenly-sampled from the adjacency matrix to approximate the expectation, and train the adjacency network each time after the adjacency matrix is updated with newly-sampled trajectories.

Although the construction of an adjacency matrix limits our method to tasks with tabular state spaces, our method can also handle continuous state spaces using goal space discretization (see our continuous control experiments in Section 5). For applications with vast state spaces, constructing a complete adjacency matrix will be problematic, but it is still possible to scale our method to these scenarios using specific feature construction or dimension reduction methods [28, 29, 7], or substituting the distance learning procedure with more accurate distance learning algorithms [10, 8] at the cost of some learning efficiency. We consider possible extensions in this direction as our future work.

### 4.2 Combining HRL and Adjacency Constraint

With a learned adjacency network $\psi_\phi$, we can now incorporate the adjacency constraint into the goal-conditioned HRL framework. According to Equation (8), we introduce an adjacency loss $\mathcal{L}_{\mathrm{adj}}$ to replace the original strict adjacency constraint and minimize the following high-level objective:

$$\mathcal{L}_{\mathrm{high}}(\theta_h) = -\mathbb{E}_{\pi_{\theta_h}^h} \sum_{t=0}^{T-1} \left( \gamma^t r_{kt}^h - \eta \cdot \mathcal{L}_{\mathrm{adj}} \right), \tag{12}$$

where $\eta$ is the balancing coefficient, and $\mathcal{L}_{\mathrm{adj}}$ is derived by replacing $d_{\mathrm{st}}$ with $\tilde{d}_{\mathrm{st}}$ defined by Equation (10) in the second term of Equation (8):

$$\mathcal{L}_{\mathrm{adj}}(\theta_h) = H \left( \tilde{d}_{\mathrm{st}} \left( s_{kt}, \varphi^{-1}(g_{kt}) | \phi \right), k \right) \propto \max \left( \|\psi_\phi(\varphi(s_{kt})) - \psi_\phi(g_{kt})\|_2 - \epsilon_k, 0 \right), \tag{13}$$

where $g_{kt} \sim \pi_{\theta_h}^h(g|s_{kt})$. Equation (13) will output a non-zero value when the generated subgoal and the current state have an Euclidean distance larger than $\epsilon_k$ in the adjacency space, indicating non-adjacency. It is thus consistent with the $k$-step adjacency constraint. In practice, we plug $\mathcal{L}_{\mathrm{adj}}$ as an extra loss term into the original policy loss term of a specific high-level RL algorithm, e.g., TD error for temporal-difference learning methods.

## 5 Experimental Evaluation

We have presented our method of Hierarchical Reinforcement learning with $k$-step Adjacency Constraint (HRAC). Our experiments are designed to answer the following questions: (1) Can HRAC promote the generation of adjacent subgoals? (2) Can HRAC improve the sample efficiency and overall performance of goal-conditioned HRL? (3) Can HRAC outperform other strategies that may also improve the learning efficiency of hierarchical agents, e.g., hindsight experience replay [1]?

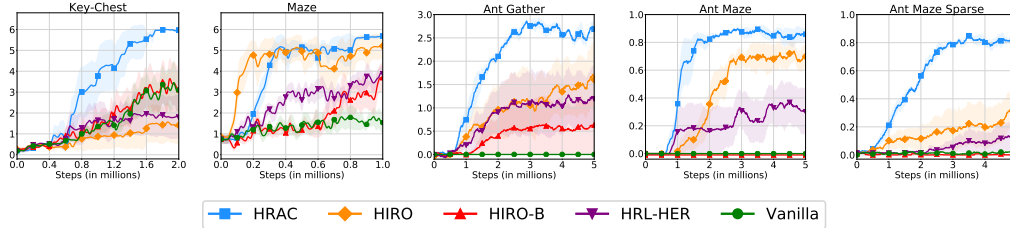

Figure 6: Learning curves of HRAC and baselines on all tasks. Each curve and its shaded region represent mean episode reward and standard error of the mean respectively, averaged over 5 independent trials. All curves have been smoothed equally for visual clarity.

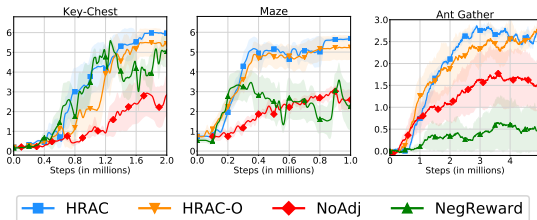

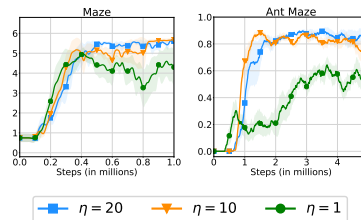

Figure 7: Learning curves in the ablation study, averaged over 5 independent trials.

Figure 8: Learning curves with different balancing coefficients.

## 5.1 Environment Setup

We employed two types of tasks with discrete and continuous state and action spaces to evaluate the effectiveness of our method, as shown in Figure 5. Discrete tasks include Key-Chest and Maze, where the agents are spawned in grid worlds with injected stochasticity and need to accomplish tasks that require both low-level control and high-level planning. Continuous tasks include Ant Gather, Ant Maze and Ant Maze Sparse, where the first two tasks are widely-used benchmarks in HRL community [6, 11, 26, 25, 22], and the third task is a more challenging navigation task with sparse rewards. In all tasks, we used a pre-defined 2-dimensional goal space that represents the $(x, y)$ position of the agent. More details of the environments are in the supplementary material.

## 5.2 Comparative Experiments

To comprehensively evaluate the performance of HRAC with different HRL implementations, we employed two different HRL instances for different tasks. On discrete tasks, we used off-policy TD3 [13] for high-level training and on-policy A2C, the syncrhonous version of A3C [24], for the low-level. On continuous tasks, we used TD3 for both the high-level and the low-level training, following prior work [26], and discretized the goal space to $1 \times 1$ grids for adjacency learning.

We compared HRAC with the following baselines. (1) *HIRO* [26]: one of the state-of-the-art goal-conditioned HRL approaches. (2) *HIRO-B*: a baseline analagous to HIRO, using binary intrinsic reward for subgoal reaching instead of the shaped reward used by HIRO. (3) *HRL-HER*: a baseline that employs hindsight experience replay (HER) [1] to produce alternative successful subgoal-reaching experiences as complementary low-level learning signals [22]. (4) *Vanilla*: Kulkarni et al. [20] used absolute subgoals instead of directional subgoals and adopted a binary intrinsic reward setting. More details of the baselines are in the supplementary material.

The learning curves of HRAC and baselines across all tasks are plotted in Figure 6. In the Maze task with dense rewards, HRAC achieves comparable performance with HIRO and outperforms other baselines, while in other tasks HRAC consistently surpasses all baselines both in sample efficiency and asymptotic performance. We note that the performance of the baseline HRL-HER matches the results in the previous study [26] where introducing hindsight techniques often degrades the performance of HRL, potentially due to the additional burden introduced on low-level training.

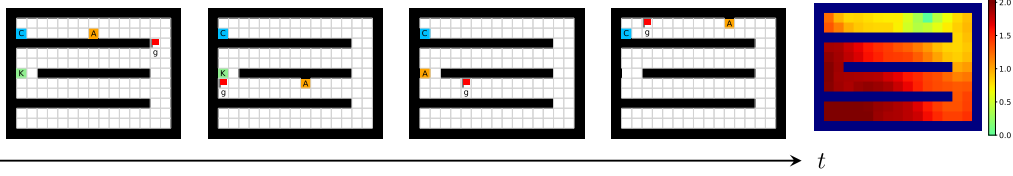

Figure 9: Visualizations on the Key-Chest task, based on a single evaluation run. The agent (A), key (K), chest (C) and subgoal (g) at four different time steps are plotted. The adjacency heatmap is based on the fourth time step, where colder colors represent smaller shortest transition distances.

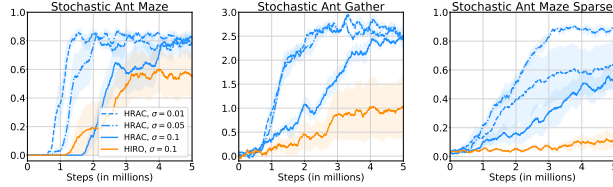

Figure 10: Learning curves in stochastic environments, averaged over 5 independent trials.

## 5.3 Ablation Study and Visualizations

We also compared HRAC with several variants to investigate the effectiveness of each component. (1) *HRAC-O*: An oracle variant that uses a perfect adjacency matrix directly obtained from the environment. We note that compared to other methods, this variant uses the additional information that is not available in many applications. (2) *NoAdj*: A variant that uses an adjacency training method analogous to the work by Savinov et al. [34, 35], where no adjacency matrix is maintained. The adjacency network is trained using state-pairs directly sampled from stored trajectories, under the same training budget as HRAC. (3) *NegReward*: This variant implements the $k$-step adjacency constraint by penalizing the high-level with a negative reward when it generates non-adjacent subgoals, which is used by HAC [22].

We provide learning curves of HRAC and these variants in Figure 7. In all tasks, HRAC yields similar performance with the oracle variant HRAC-O while surpassing the NoAdj variant by a large margin, exhibiting the effectiveness of our adjacency learning method. Meanwhile, HRAC achieves better performance than the NegReward variant, suggesting the superiority of implementing the adjacency constraint using a differentiable adjacency loss, which provides a stronger supervision than a penalty. We also empirically studied the effect of different balancing coefficients $\eta$. Results are shown in Figure 8, which suggests that generally a large $\eta$ can lead to better and more stable performance.

Finally, we visualize the subgoals generated by the high-level policy and the adjacency heatmap in Figure 9. Visualizations indicate that the agent does learn to generate adjacent and interpretable subgoals. We provide additional visualizations in the supplementary material.

## 5.4 Empirical Study in Stochastic Environments

To empirically verify the stochasticity robustness of HRAC, we applied it to a set of stochastic tasks, including stochastic Ant Gather, Ant Maze and Ant Maze Sparse tasks, which are modified from the original ant tasks respectively. Concretely, we added Gaussian noise with different standard deviations $\sigma$ to the $(x, y)$ position of the ant robot at every step, including $\sigma = 0.01$, $\sigma = 0.05$ and $\sigma = 0.1$, representing increasing environmental stochasticity. In these tasks we compare HRAC with the baseline HIRO, which has exhibited generally better performance than other baselines, in the most noisy scenario when $\sigma = 0.1$. As displayed in Figure 10, HRAC achieves similar asymptotic performances with different noise magnitudes in stochastic Ant Gather and Ant Maze tasks and consistently outperforms HIRO, exhibiting robustness to stochastic environments.

# 6 Related Work

Effectively learning policies with multiple hierarchies has been a long-standing problem in RL. Goal-conditioned HRL [5, 37, 20, 42, 26, 22] aims to resolve this problem with a framework that separates high-level planning and low-level control using subgoals. Recent advances in goal-conditioned HRL mainly focus on improving the learning efficiency of this framework. Nachum et al. [26, 25] proposed an off-policy correction technique to stabilize training, and addressed the problem of goal space representation learning using a mutual-information-based objective. However, the subgoal generation process in their approaches is unconstrained and supervised only by the external reward, and thus these methods may still suffer from training inefficiency. Levy et al. [22] used hindsight techniques [1] to train multi-level policies in parallel and also penalized the high-level for generating subgoals that the low-level failed to reach. However, their method has no theoretical guarantee, and they directly obtain the reachability measure from the environment, using the environmental information that is not available in many scenarios. There is also prior work focusing on unsupervised acquisition of subgoals based on potentially pivotal states [23, 18, 21, 34, 32, 17]. However, these subgoals are not guaranteed to be well-aligned with the downstream tasks and thus are often sub-optimal.

Several prior works have constructed an environmental graph for high-level planning used search nearby graph nodes as reachable subgoals for the low-level [34, 8, 17, 45]. However, these approaches hard-coded the high-level planning process based on domain-specific knowledge, e.g., treat the planning process as solving a shortest-path problem in the graph instead of a learning problem, and thus are limited in scalability. Nasiriany et al. [29] used goal-conditioned value functions to measure the feasibility of subgoals, but a pre-trained goal-conditioned policy is required. A more general topic of goal generation in RL has also been studied in the literature [12, 28, 33]. However, these methods only have a flat architecture and therefore cannot successfully solve tasks that require complex high-level planning.

Meanwhile, our method relates to previous research that studied transition distance or reachability [30, 34, 35, 10, 15]. Most of these works learn the transition distance based on RL [30, 10, 15], which tend to have a high learning cost. Savinov et al. [34, 35] proposed a supervised learning approach for reachability learning. However, the metric they learned depends on a certain policy used for interaction and thus could be sub-optimal compared to our learning method. There are also other metrics that can reflect state similarities in MDPs, such as successor representation [4, 21] that depends on both the environmental dynamics and a specific policy, and bisimulation metrics [9, 3] that depend on both the dynamics and the rewards. Compared to these metrics, the shortest transition distance depends only on the dynamics and therefore may be seamlessly applied to multi-task settings.

# 7 Conclusion

We present a novel $k$-step adjacency constraint for goal-conditioned HRL framework to mitigate the issue of training inefficiency, with the theoretical guarantee of preserving the optimal policy in deterministic MDPs. We show that the proposed adjacency constraint can be practically implemented with an adjacency network. Experiments on several testbeds with discrete and continuous state and action spaces demonstrate the effectiveness and robustness of our method.

As one of the most promising directions for scaling up RL, goal-conditioned HRL provides an appealing paradigm for handling large-scale problems. However, some key issues involving how to devise effective and interpretable hierarchies remain to be solved. For instance, the hierarchical structure may endow the high-level policy with the ability to learn and explore in a more semantically meaningful space [27], and the subgoals may be shared and reused in multi-task settings. Other future work may include extending our method to tasks with high-dimensional state spaces, e.g., by encompassing modern representation learning schemes [16, 25, 38], and leveraging the learned adjacency network to improve learning efficiency in more general scenarios.

## Broader Impact

This work may promote the research in the field of HRL and RL, and has potential real-world applications such as robotics. The main uncertainty of the proposed method might be the fact that the RL training process itself is somewhat brittle, and may break in counterintuitive ways when the

reward function is misspecified. Also, since the training data of RL heavily depends on the training environments, designing unbiased simulators or real-world training environments is important for eliminating the biases in the data collected by the agents.

## Acknowledgments and Disclosure of Funding

This work was supported in part by the National Natural Science Foundation of China under Grant 61671266, Grant 61836004, Grant 61836014 and in part by the Tsinghua-Guoqiang research program under Grant 2019GQG0006. The authors would also like to thank the anonymous reviewers for their careful reading and their many insightful comments.

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
