[Supplementary Material]

# A Proofs of Theorems

## A.1 Proof of Theorem 1

*Proof.* Under the assumption that the MDP is deterministic and all states are strongly connected, there exists at least one shortest state trajectory from $s$ to $g$. Without loss of generality, we consider one shortest state trajectory $\tau^* = (s_0, s_1, s_2, \cdots, s_{n-1}, s_n)$, where $s_0 = s$, $s_n = \varphi^{-1}(g)$ and $d_{st}\left(s, \varphi^{-1}(g)\right) = n$. For all $k \in \mathbb{N}_+$ and $k \leq d_{st}\left(s, \varphi^{-1}(g)\right) = n$, let $\tilde{g} = \varphi(s_k)$, and let $\tau = (s_0, s_1, s_2, \cdots, s_k)$ be the $k$-step sub-trajectory of $\tau^*$ from $s_0$ to $s_k$. Since $s_0$ and $s_k$ is connected by $\tau$ in $k$ steps, we have that $d_{st}\left(s_0, \varphi^{-1}(\tilde{g})\right) = d_{st}(s_0, s_k) \leq k$, i.e., $\tilde{g} \in \mathcal{G}_A(s, k)$. In the following, we will prove that $\pi^*(s_i, \tilde{g}) = \pi^*(s_i, g)$, $\forall s_i \in \tau$ $(i \neq k)$.

We first prove that the shortest transition distance $d_{st}$ satisfies the triangle inequality, i.e., consider three arbitrary states $s_1, s_2, s_3 \in \mathcal{S}$, then $d_{st}(s_1, s_3) \leq d_{st}(s_1, s_2) + d_{st}(s_2, s_3)$: let $\tau_{12}^*$ be one shortest state trajectory between $s_1$ and $s_2$ and let $\tau_{23}^*$ be one shortest state trajectory between $s_2$ and $s_3$. We can concatenate $\tau_{12}^*$ and $\tau_{23}^*$ to form a trajectory $\tau_{13} = (\tau_{12}^*, \tau_{23}^*)$ that connects $s_1$ and $s_3$. Then, by Definition 1 we have $d_{st}(s_1, s_3) \leq d_{st}(s_1, s_2) + d_{st}(s_2, s_3)$.

Using the triangle inequality, we can prove that the sub-trajectory $\tau = (s_0, s_1, s_2, \cdots, s_k)$ is also a shortest trajectory from $s_0 = s$ to $s_k$: assume that this is not true and there exists a shorter trajectory from $s_0$ to $s_k$. Then, by Definition 1 we have $d_{st}(s_0, s_k) < k$. Since $(s_k, s_{k+1}, \cdots, s_n)$ is a valid trajectory from $s_k$ to $s_n$, we have $d_{st}(s_0, s_k) \leq n - k$. Applying the triangle inequality, we have $d_{st}(s_0, s_n) \leq d_{st}(s_0, s_k) + d_{st}(s_k, s_n) < k + n - k = n$, which is in contradiction with $d_{st}\left(s, \varphi^{-1}(g)\right) = d_{st}(s_0, s_n) = n$. Thus, our original assumption must be false, and the trajectory $\tau = (s_0, s_1, s_2, \cdots, s_k)$ is a shortest trajectory from $s_0$ to $s_k$.

Finally, let $\alpha : \mathcal{S} \times \mathcal{S} \rightarrow \mathcal{A}$ be an inverse dynamics model, i.e., given state $s_t$ and the next state $s_{t+1}$, $\alpha(s_t, s_{t+1})$ outputs the action $a_t$ that is performed at $s_t$ to reach $s_{t+1}$. Then, employing Equation (3), for $i = 0, 1, \cdots, k-1$ we have $\pi^*(s_i, g) = \alpha(s_i, s_{i+1})$ given that $\tau^*$ is a shortest trajectory from $s_0$ to $\varphi^{-1}(g)$, and $\pi^*(s_i, \tilde{g}) = \alpha(s_i, s_{i+1})$ given that $\tau$ is a shortest trajectory from $s_0$ to $\varphi^{-1}(\tilde{g})$. This indicates that $\pi^*(s_i, \tilde{g}) = \pi^*(s_i, g)$, $\forall s_i \in \tau$ $(i \neq k)$. $\qquad \square$

## A.2 Proof of Theorem 2

*Proof.* Using Theorem 1, we have that for each subgoal $g_{kt}$, $t = 0, 1, \cdots, T - 1$, there exists a subgoal $\tilde{g}_{kt} \in \mathcal{G}_A(s_{kt}, k)$ that can induce the same low-level $k$-step action sequence as $g_{kt}$. This indicates that the agent's trajectory and the high-level reward $r_{kt}^h$ defined by Equation (1) remain the same for all $t$ when replacing $g_{kt}$ with $\tilde{g}_{kt}$. Then, using the high-level Bellman optimality equation for the optimal $Q$ function

$$\begin{aligned} Q^*(s_{kt}, g_{kt}) &= r_{kt}^h + \gamma \max_{g \in \mathcal{G}} Q^*(s_{k(t+1)}, g) \\ &= r_{kt}^h + \gamma Q^*(s_{k(t+1)}, g_{k(t+1)}), \quad t = 0, 1 \cdots, T - 1 \end{aligned} \tag{14}$$

and $Q^*(s_{kT}, g) = 0$, $\forall g \in \mathcal{G}$ as $s_{kT}$ is the final state of $\tau^*$, we have $Q^*(s_{kt}, \tilde{g}_{kt}) = Q^*(s_{kt}, g_{kt})$, $t = 0, 1, \cdots, T - 1$. $\qquad \square$

# B Implementation Details

## B.1 Adjacency Learning

**Constructing and updating the adjacency matrix.** We use the agent's trajectories to construct and update the adjacency matrix. Concretely, the adjacency matrix is initialized to an empty matrix at the beginning of training. Each time when the agent explores a new state that it has never visited before, the adjacency matrix is augmented by a new row and a new column with zero elements, representing the $k$-step adjacent relation between the new state and explored states. When the temporal distance between two states in one trajectory is not larger than $k$, then the corresponding element in the adjacency matrix will be labeled to 1, indicating the adjacency. (The diagnoal of the adjacency matrix will always be labeled to 1.) Although the temporal distance between two states based on a single trajectory is often larger than the real shortest transition distance, it can

Figure 11: Qualitative comparison of adjacency learning methods. (a) Environment layout. The agent starts from the grid A. (b) Results of our method, including the adjacency heatmaps from states $s_1$, $s_2$ and the LLE visualization of state embeddings. (c) Results of the method proposed by Savinov et al. [34, 35], including the adjacency heatmaps from states $s_1$, $s_2$ and the LLE visualization of state embeddings.

be easily shown that the adjacency matrix with this labeling strategy can converge to the optimal adjacency matrix asymptotically with sufficient trajectories sampled by different policies. In practice, we employ a trajectory buffer to store newly-sampled trajectories, and update the adjacency matrix online in a fixed frequency using the stored trajectories. The trajectory buffer is cleared after each update.

**Training the adjacency network.** The adjacency network is trained by minimizing the objective defined by Equation (11). We use states evenly-sampled from the adjacency matrix (i.e. from the set of all explored states) to approximate the expectation, and train the adjacency network each time after the adjacency matrix is updated with new trajectories. Note that by explicitly aggregating the adjacency information using an adjacency matrix, we are able to achieve the uniform sampling of all explored states and thus achieve a nearly unbiased estimation of the expectation, which cannot be realized when we directly sample state-pairs from the trajectories (see the following comparison with the work by Savinov et al. [34, 35] for details).

Embedding all subgoals with a single adjacency network is enough to express adjacency when the environment is reversible. However, when this condition is not satisfied, it is insufficient to express directional adjacency using one adjacency network, as the parameterized approximation defined by Equation (10) is symmetric for $s_1$ and $s_2$. In this case, one can use two separate sub-networks to embed $g_1$ and $g_2$ in Equation (10) respectively using the structure proposed in UVFA [36].

**Comparison with the work by Savinov et al.** Savinov et al. [34, 35] also propose a supervised learning approach for learning the adjacency between states. The main differences between our method and theirs are: 1) We use trajectories sampled by multiple policies to construct training samples, while they only use trajectories sampled by one specific policy; 2) We use an adjacency matrix to explicitly aggregate the adjacency information and sample training pairs based on the adjacency matrix, while they directly sample training pairs from trajectories. These differences lead to two advantages of our method: 1) By using multiple policies, we achieve a more accurate adjacency approximation, as shown by Equation (9); 2) By maintaining an adjacency matrix, we can uniformly sample from the set of all explored states and realize a nearly unbiased estimation of the expectation in Equation (11), while the estimation by sampling state-pairs from trajectories is biased. As an example, consider a simple grid world in Figure 11(a), where states are represented by their $(x, y)$ positions. In this environment, states $s_1$ and $s_2$ are non-adjacent since they are separated by a wall. However, it is hard for the method by Savinov et al. to handle this situation as these two

---

**Algorithm 1** HRAC

---

**Input:** High-level policy $\pi_{\theta_h}^h$ parameterized by $\theta_h$, low-level policy $\pi_{\theta_l}^l$ parameterized by $\theta_l$, adjacency network $\psi_\phi$ parameterized by $\phi$, state-goal mapping function $\varphi$, goal transition function $h$, high-level action frequency $k$, number of training episodes $N$, adjacency learning frequency $C$, empty adjacency matrix $\mathcal{M}$, empty trajectory buffer $\mathcal{B}$.

Sample and store trajectories in the trajectory buffer $\mathcal{B}$ using a random policy.
Construct the adjacency matrix $\mathcal{M}$ using the trajectory buffer $\mathcal{B}$.
Pre-train $\psi_\phi$ using $\mathcal{M}$ by minimizing Equation (11).
Clear $\mathcal{B}$.
**for** $n = 1$ **to** $N$ **do**
    Reset the environment and sample the initial state $s_0$.
    $t = 0$.
    **repeat**
        **if** $t \equiv 0 \,(\mathrm{mod}\, k)$ **then**
            Sample subgoal $g_t \sim \pi_{\theta_h}^h(g|s_t)$.
        **else**
            Perform subgoal transition $g_t = h(g_{t-1}, s_{t-1}, s_t)$.
        **end if**
        Sample low-level action $a_t \sim \pi_{\theta_l}^l(a|s_t, g_t)$.
        Sample next state $s_{t+1} \sim \mathcal{P}(s|s_t, a_t)$.
        Sample reward $r_t \sim \mathcal{R}(r|s_t, a_t)$.
        Sample episode end signal *done*.
        $t = t + 1$.
    **until** *done* is *true*.
    Store the sampled trajectory in $\mathcal{B}$.
    Train high-level policy $\pi_{\theta_h}^h$ according to Equation (12) and (13).
    Train low-level policy $\pi_{\theta_l}^l$.
    **if** $n \equiv 0 \,(\mathrm{mod}\, C)$ **then**
        Update the adjacency matrix $\mathcal{M}$ using the trajectory buffer $\mathcal{B}$.
        Fine-tune $\psi_\phi$ using $\mathcal{M}$ by minimizing Equation (11).
        Clear $\mathcal{B}$.
    **end if**
**end for**

---

states rarely emerge in the same trajectory due to the large distance, and thus the loss induced by this state-pair is very likely to be dominated by the loss of other nearer state-pairs. Meanwhile, our method treat the loss of all state-pairs equally, and can therefore alleviate this phenomenon. Empirically, we employed a random agent (since the random policy is stochastic, it can be viewed as multiple deterministic policies, and is enough for adjacency learning in this simple environment) to interact with the environment for $20,000$ steps, and trained the adjacency network with collected samples using both methods. We visualize the LLE of state embeddings and two adjacency distance heatmaps by both methods respectively in Figure 11(b) and 11(c). Visualizations validate our analysis, showing that our method does learn a better adjacency measure in this scenario.

## B.2 Algorithm Pseudocode

We provide Algorithm 1 to show the training procedure of HRAC. Some training details are omitted for brevity, e.g. the detailed training process of the low-level policy.

## B.3 Environment Details

**Maze.** This environment has a size of $13 \times 17$, with a discrete 2-dimensional state space representing the $(x, y)$ position of the agent and a discrete 4-dimensional action space corresponding to actions moving towards four directions. The agent is provided with a dense reward to facilitate exploration, i.e., $+0.1$ each step if the agent moves closer to the goal, and $-0.1$ each step if the agent moves

farther. Each episode has a maximum length of 200. Environmental stochasticity is introduced by replacing the action of the agent by a random action each step with a probability of 0.25.

**Key-Chest.** This environment has a size of $13 \times 17$, with a discrete 3-dimensional state space in which the first two dimensions represent the $(x, y)$ position of the agent respectively, and the third dimension represents whether the agent has picked up the key (1 if the agent has the key and 0 otherwise). The agent has the same action space as the Maze task. The agent is provided with sparse reward of $+1$ and $+5$, respectively for picking up the key and opening the chest. Each episode ends if the agent opens the chest or runs out of the step limit of 500. The random action probability of the environment is also 0.25.

**Ant Gather.** This environment has a size of $20 \times 20$, with a continuous state space including the current position and velocity, the current time step $t$, and the depth readings defined by the stardard Gather environment [6]. We use the ant robot pre-defined by Rllab, with a 8-dimensional continuous action space. The ant robot is spawned at the center of the map and needs to gather apples while avoiding bombs. Both apples and bombs are randomly placed in the environment at the beginning of each episode. The agent receives a positive reward of $+1$ for each apple and a negative reward of $-1$ for each bomb. Each episode terminates at 500 time steps.

**Ant Maze.** This environment has a size of $24 \times 24$, with a continuous state space including the current position and velocity, the target location, and the current time step $t$. In the training stage, the environment randomly samples a target position at the beginning of each episode, and the agent receives a dense reward at each time step according to its negative Euclidean distance from the target position. At evaluation stage, the target position is fixed to $(0, 16)$, and the success is defined as being within an Euclidean distance of 5 from the target. Each episode ends at 500 time steps. In practice, we scale the environmental reward by 0.1 equally for all methods.

**Ant Maze Sparse.** This environment has a size of $20 \times 20$, with the same state and action spaces as the Ant Maze task. The target position (goal) is set at the position $(2.0, 9.0)$ in the center corridor. The agent is rewarded by $+1$ only if it reachs the goal, which is defined as having a Euclidean distance that is smaller than 1 from the goal. At the beginning of each episode, the agent is randomly placed in the maze except at the goal position. Each episode is terminated if the agent reaches the goal or after 500 steps.

## B.4 HRAC and Baseline Details

We use PyTorch to implement our method HRAC and all the baselines.[2]

**HRAC.** For discrete control tasks, we adopt a binary intrinsic reward setting: we set the intrinsic reward to 1 when $|s_x - g_x| \leq 0.5$ and $|s_y - g_y| \leq 0.5$, where $(s_x, s_y)$ is the position of the agent and $(g_x, g_y)$ is the position of the desired subgoal. For continuous control tasks, we adopt a dense intrinsic reward setting based on the negative Euclidean distances $-\|s - g\|_2$ between states and subgoals.

**HIRO.** Following Nachum et al. [26], we restrict the output of high-level to $(\pm 10, \pm 10)$, representing the desired shift of the agent's $(x, y)$ position. By limiting the range of directional subgoals generated by the high-level, HIRO can roughly control the Euclidean distance between the absolute subgoal and the current state in the raw goal space rather than the learned adjacency space.

**HRL-HER.** As HER cannot be applied to the on-policy training scheme in a straightforward manner, in discrete control tasks where the low level policy is trained using A2C, we modify its implementation so that it can be incorporated into the on-policy setting. For this on-policy variant, during the training phase, we maintain an additional episodic state memory. This memory stores states that the agent has visited from the beginning of each episode. When the high-level generates a new subgoal, the agent randomly samples a subgoal mapped from a stored state with a fixed probability 0.2 to substitute the generated subgoal for the low-level to reach. This implementation resembles

the "episode" strategy introduced in the original HER. We still use the original HER in continuous control tasks.

**NoAdj.** We follow the training pipeline proposed by Savinov et al. [34, 35], where no adjacency matrix is maintained. Training pairs are constructed by randomly sampling state-pairs $(s_i, s_j)$ from the stored trajectories. The samples with $|i - j| \leq k$ are labeled as positive with $l = 1$, and the samples with $|i - j| \geq Mk$ are negative ones with $l = 0$. The hyper-parameter $M$ is used to create a gap between the two types of samples, where in practice we use $M = 4$.

**NegReward.** In this variant, every time the high-level generates a subgoal, we use the adjacency network to judge whether it is $k$-step adjacent. If the subgoal is non-adjacent, the high-level will be penalized with a negative reward $-1$.

### B.5 Network Architecture

For the hierarchical policy network, we employ the same architecture as HIRO [26] in continuous control tasks, where both the high-level and the low-level use TD3 [13] algorithm for training. In discrete control tasks, we use two networks consisting of 3 fully-connected layers with ReLU nonlinearities as the low-level actor and critic networks of A2C (our preliminary results show that the performances using on-policy and off-policy methods for the low-level training are similar in the discrete control tasks we consider), and use the same high-level TD3 network architecture as the continuous control task. The size of the hidden layers of both low-level actor and critic is $(300, 300)$. The output of high-level actor is activated using the `tanh` function and scaled to fit the size of the environments.

For the adjacency network, we use a network consisting of 4 fully-connected layers with ReLU nonlinearities in all tasks. Each hidden layer of the adjacency network has the size of $(128, 128)$. The dimension of the output embedding is 32.

We use Adam optimizer for all networks.

### B.6 Hyper-parameters

We list all hyper-parameters we use in the discrete and continuous control tasks respectively in Table 1 and Table 2, and list the hyper-parameters used for adjacency network training in Table 3. "Ranges" in the tables show the ranges of hyper-parameters considered, and the hyper-parameters without ranges are not tuned.

## C   Additional Visualizations

We provide additional subgoal and adjacency heatmap visualizations of the Maze and Key-Chest tasks respectively in Figure 12 and Figure 13.

Table 1: Hyper-parameters used in discrete control tasks. "K-C" in the table refers to "Key-Chest".

| Hyper-parameters | Values | Ranges |
|---|:---:|:---:|
| High-level TD3 | | |
| Actor learning rate | 0.0001 | |
| Critic learning rate | 0.001 | |
| Replay buffer size | 10000 / 20000 for Maze / K-C | {10000, 20000} |
| Batch size | 64 | |
| Soft update rate | 0.001 | |
| Policy update frequency | 2 | {1, 2} |
| $\gamma$ | 0.99 | |
| High-level action frequency $k$ | 10 | |
| Reward scaling | 1.0 | |
| Exploration strategy | Gaussian ($\sigma = 3.0/5.0$ for Maze / K-C) | {3.0, 5.0} |
| Adjacency loss coefficient $\eta$ | 20 | {1, 5, 10, 20} |
| Low-level A2C | | |
| Actor learning rate | 0.0001 | |
| Critic learning rate | 0.0001 | |
| Entropy weight | 0.01 | |
| $\gamma$ | 0.99 | |
| Reward scaling | 1.0 | |

Table 2: hyper-parameters used in continuous control tasks.

| Hyper-parameters | Values | Ranges |
|---|:---:|:---:|
| High-level TD3 | | |
| Actor learning rate | 0.0001 | |
| Critic learning rate | 0.001 | |
| Replay buffer size | 200000 | |
| Batch size | 128 | |
| Soft update rate | 0.005 | |
| Policy update frequency | 1 | |
| $\gamma$ | 0.99 | |
| High-level action frequency $k$ | 10 | |
| Reward scaling | 0.1 / 1.0 for Ant Maze / others | {0.1, 1.0} |
| Exploration strategy | Gaussian ($\sigma = 1.0$) | {1.0, 2.0} |
| Adjacency loss coefficient $\eta$ | 20 | {1, 5, 10, 20} |
| Low-level TD3 | | |
| Actor learning rate | 0.0001 | |
| Critic learning rate | 0.001 | |
| Replay buffer size | 200000 | |
| Batch size | 128 | |
| Soft update rate | 0.005 | |
| Policy update frequency | 1 | |
| $\gamma$ | 0.95 | |
| Reward scaling | 1.0 | |
| Exploration strategy | Gaussian ($\sigma = 1.0$) | |

Table 3: Hyper-parameters used in adjacency network training.

| Hyper-parameters | Values | Ranges |
|---|---|---|
| **Adjacency Network** | | |
| Learning rate | 0.0002 | |
| Batch size | 64 | |
| $\epsilon_k$ | 1.0 | |
| $\delta$ | 0.2 | |
| Steps for pre-training | 50000 | |
| Pre-training epochs | 50 | |
| Online training frequency (steps) | 50000 | |
| Online training epochs | 25 | |

Figure 12: Additional subgoal and adjacency heatmap visualizations of the Maze task, based on a single evaluation run. The agent (A), goal (G) and subgoal (g) at different time steps in one episode are plotted. Colder colors in the adjacency heatmaps represent smaller shortest transition distances.

Figure 13: Additional subgoal and adjacency heatmap visualizations of the Key-Chest task, based on a single evaluation run. The agent (A), key (K), chest (C) and subgoal (g) at different time steps in one episode are plotted. Colder colors in the adjacency heatmaps represent smaller shortest transition distances.

## Footnotes

[2]We use the open source PyTorch implementation of HIRO at `https://github.com/bhairavmehta95/data-efficient-hrl`.