[Reviews · NeurIPS 2020]

Review 1

Summary and Contributions: The paper provides an interesting method for constraining the set of goals considered by a high level policy within the context of hierarchical agents. Although many such agents have been able to provide interesting results by training a high level policy that selects goals and a low level policy that achieves those goals, it many cases the space of all possible goals is much too complex both from the point of view of the high level policy, which might have to make choices over a very large set of goals, as well as the low level policy, which might have to be trained with sparse signals due to selection of goals that are not easily achievable. The paper provides straight forward yet interesting theoretical results on the ability to maintain optimality even when the set of goals is reduced to those that can be achieved in a fixed finite number of steps. Then the authors provide both an approximation to the proposed goal space reduction and a careful empirical analysis that compares the proposed technique to similar algorithms that have been proved to generate SOTA results in a number of OpenAI Gym tasks.

Strengths: The approach has a very neat theoretical motivation and the paper was carefully presented and properly contextualized within the latest advances in Hierarchical Reinforcement Learning with goal conditioning. The work has strong relevance for the NeurIPS community, as hierarchical methods are one of the most promising approaches to scaling RL; facilitating training of higher level policies for better goal conditioning is an important challenge.

Weaknesses: Although the paper is well motivated and properly contextualized, the results are mostly incremental. The theory is derived in the context of deterministic MDPs, and practical implementations of the ideas presented in the paper are quite distant from the theory and the initial motivation. The authors argue that the deterministic assumption is not a strong one and that "many real-world applications can be approximated as environments with deterministic dynamics". Although I am in agreement with the authors with respect to the tasks demonstrated in the empirical section, I don't think this is a strong argument, nor that this statement is properly supported in this paper or in the cited paper (see reference [11] in line 135). Many interesting RL tasks are both stochastic and partially observable and I can definitely see many issues with the approach presented in this paper when stochasticity and partial observability are combined. On the novelty of the approach: the paper provides a number of very good references for reducing the goal space and I think the approach presented here is very similar to (1) methods where the high level policy has to decide over reduced deltas in the some hidden representation of the state, and (2) methods where goals match notions of short term changes (based on the dynamics of the environment. The approximation of the k-step adjacency matrix is interesting, but of limited novelty.

Correctness: The paper is generally very well presented and the clarity of the presentation makes the presentation quite intuitive and easy to follow. The claims are correct, as far as I can tell, and the empirical work is complete with ablations, comparative analyses to other strong similar methods, and nicely visualized. Still the authors make two claims that I would like to challenge: (1) "Searching in a large goal space poses difficulties for both high-level subgoal generation and low-level policy learning. In this paper, we show that this problem can be effectively alleviated by restricting the high-level action space from the whole goal space to a k-step adjacency region centered by the current state using an adjacency constraint" - The authors provide great references to methods where the goal space can be easily interpreted as "short term changes" which is equivalent to the k-step adjacency. I have a hard time understanding the reduction in goals considered by the high-level policy, when compared to these local methods. I think the paper needs a more careful treatise of the impact of the proposed "goal reduction" on sets of goals that are already reduced due to some more general constraints (e.g. incremental change in one feature). (2) "Experimental results on discrete and continuous control tasks show that our method outperforms state-of-the-art HRL approaches." Since the empirical section are based only on comparisons to HIRO and a variation of HER, I think this claim extrapolates the results. I think the gains over HIRO are great and make the paper quite valuable already. Since the algorithm is an incremental change on the approach that HIRO takes in selecting goals, I would refrain from claiming that "the method outperforms SOTA HRL approaches" and instead go for "the constraint was implemented to improve the performance of existing SOTA methods."

Clarity: With the exception of a few minor changes, the paper is very well written and a joy to read. L13: "outperforms the state-of-the-art HRL approaches" -> "outperforms state-of-the-art HRL approaches". L21: "which stills remains" -> which still remains" L42: "higher hierarchy"? consider proof-reading L123: Please provide a clear definition of \pi^{h*}. I don't think this has a definition that is as obvious as the low-level optimality criteria. Since the theoretical results are all claims on this optimal policy, please make sure that readers get a very good grasp of the concept. L127: I would stick to the concept of "high-level goal space" instead of "high-level action space". I believe it is quite important to distinguish the methods discussed in this paper from other methods where low level policies are not goal-conditioned. Moreover, it will improve the consistency throughout the paper. Page 4: Although you mention the deterministic MDP assumption in Theorem 1, I was confused whether the rest of the claims are also based on the same assumption or not. I think it would help if you would be upfront about this rather strong assumption in the abstract, in the introduction, and in the beginning of section 3. L193: "Although the construction of an adjacent matrix limits our method to tasks with finite states". maybe instead of "finite" the statement should be more on "tabular" state spaces. I can think of many finite state space MDPs where this method would be intractable, and the idea of goal space discretization can be applied both to large finite and continuous state spaces. L213: I believe question number (3) could be a bit more clear. "other strategies that also aim to improve learning efficiency of HRL" is a rather vague description of HER, and "hindsight technique" is hardly appropriate. Please consider a more thoughtful description of the comparison with HER. L299: "the robotics" -> "robotics"

Relation to Prior Work: As I mentioned before, the work is generally very well contextualized and relation to prior work is well presented. Still, I would like the authors to provide a more detailed comparison to the following recent works that, IMO are quite relevant: Castro, Pablo. Scalable methods for computing state similarity in deterministic Markov Decision Processes https://arxiv.org/abs/1911.09291 Khetarpal et al. What can I do here? A Theory of Affordances in Reinforcement Learning https://arxiv.org/abs/2006.15085

Reproducibility: Yes

Additional Feedback: ----- After author's response----- I would like to thank the authors for their detailed responses to reviewer's concerns. Please make sure to include all of the interesting distinctions to related work that you provided in the rebuttal. My score stands at a strong 8. The paper is very interesting, relevant, and well written. I still believe that the analysis being done My overall score still stands at a strong 8. I still stand my by claim that deriving the theory on deterministic MDPs is the main weakness of the paper. As mentioned before, the empirical results are based on environments where one could indeed argue that they can be approximated as environments with deterministic dynamics, but my skepticism is mostly related to environments where this is not the case.


Review 2

Summary and Contributions: In this work, the authors propose a goal-based HRL-framework that works on two-levels, a meta or high-level policy that generates the sub-goals that are at most k-steps away, and a low-level policy that is that learns how to reach to the conditioned sub-goal. They show that in deterministic MDPs their HRL method can achieve the optimal solution. The k-step adjacent goals generation constraint is included as a regularization objective along with the k-step rewards in the high-level policy and low-level policy is trained using the intrinsic reward that is based on closeness to sub-goal metric. The adjacency constraint is implemented in practice via a novel adjacency network-based approach for learning the notion of closeness in MDPs.

Strengths: - A very well written work with strong empirical results that highlight the effectiveness of their methods against the other baselines. - Novel distance metric learning approach for MDPs based on contrastive loss that seems to work well even for the continuous Mujoco tasks.

Weaknesses: - The theoretical results are limited to deterministic MDPs. If I understand correctly the distance metric training is also limited to deterministic policies (Eq 9). While the authors have addressed this point a little, this limits the work by a lot and it'll be nice to have some description of whether the baselines also suffer from such assumptions. - It would have been nice if the authors added a bit more details on how their approach compares to traditional methods that are suited for learning distance metrics for MDPs such as EigenOptions [1] Ref: [1] Eigenoption Discovery through the Deep Successor Representation, Machado et al 2017

Correctness: The works seems technically sound.

Clarity: The paper is written generally in a very clear manner.

Relation to Prior Work: Yes.

Reproducibility: Yes

Additional Feedback: I hope the authors can clarify the following points; - How does this differ from say when Options of at most k-length? Aren't they equivalent? What is the advantage of your approach then? - For the discrete tasks, why is off-policy and on-policy training methods are used, whereas for the continuous only on-policy methodology is followed.


Review 3

Summary and Contributions: This paper aims to improve the sub-goal generation process in goal-conditioned hierarchical RL (HRL). The authors argue that one of the main bottlenecks in HRL is the quality of sub-goal generation. The main idea is penalizing the high-level policy when it proposes a hard-to-reach (i.e., non-k-reachable with certain parameter k) sub-goal. This idea is implemented by learning an adjacency network (i.e., reachability network [Savinov, 2019]) that predicts the # minimum transition steps required to reach from one state to another. The authors propose to bootstrap the learning of the adjacency network from the adjacency matrix. The theoretical result proves that the k-reachable constraint on high-level policy does not hurt the optimality of the learned policy. The experiment was conducted on several simple domains, and shows that the proposed method outperforms all the baselines. The ablation study demonstrates that each of the proposed components is significantly contributing to the final performance.

Strengths: The main idea is simple, interesting, and well-motivated. The empirical evaluation is made on standard goal-conditioned RL tasks such as AntMaze, where the proposed method outperforms the previous works The theoretical result supports the validity of the proposed constraint. Overall, the paper is well-written and formulation is sound.

Weaknesses: The novelty is limited (see the detailed comments below). Important baselines are missing (see the detailed comments below).

Correctness: High. The claim and method is simple and clear, and well-supported by the empirical result and theory.

Clarity: High. The paper reads well

Relation to Prior Work: Medium. I have a concern that it might be not properly crediting several prior works based on landmark (or sub-goal)-graphs and the reachability network. (See detailed comments)

Reproducibility: Yes

Additional Feedback: Overall, the paper is quite well-written and the motivation and idea are simple and interesting. Except for the two main concerns that I will describe below, I’m mostly satisfied with the quality of this paper; thus, I’m willing to increase my score if the authors can address the following concerns. 1. Limited contribution (+ possibly under-credited related works) Main point: The main idea (or contribution) of this paper is 1) constraining the generated sub-goal to be near the current state, and 2) doing so via k-step binary-adjacency. Both of these ideas have been proposed and used in several previous HRL works; thus, this can be seen as a combination of two existing ideas. 1) Constraining the generated sub-goal to be near the current state. This idea has been already used in [1-4], even though some of these works in different settings. For example, they predict the “distance” between current state and the sub-goal state (e.g., UVF with -1 step reward [1, 2] or success rate of (random) low-level policy [3], or k-step reachability [4]), and 2) limit the sub-goal generation to choose the near-by subgoals only (e.g., thresholding [1-4]). Especially, [4] uses the reachability distance which is functionally identical to the adjacency distance (Eq.(10) and (13)) used in this paper. 2) Adjacency network First, the adjacency network and reachability network [4, 5] are identical. The only additional part in this paper is bootstrapping the adjacency network learning with adjacency matrix. However, as pointed out in the paper, constructing an adjacency matrix is possible only for deterministic MDP with finite and small-sized state and action space. In this regard, it would be great if authors can empirically show how robust the adjacency matrix is in case of stochastic MDP with large state and action space; the stochasticity implemented in “maze” task is too local and the state and action space was relatively small. 2. Missing baselines As pointed out in the above, there are several related works that are closely related to this idea. Since the main contribution of this paper is improving sub-goal generation, I believe it should be compared with other subgoal generation methods used in [1-4]. ----- After author's response----- 1. Stochasticity The stochastic AntMaze experiment result was quite helpful for me to lessen my concerns on the practical performance of this method under reasonable stochasticity. I recommend authors to include the result in the appendix. 2. Comparison with graph-based methods I agree that [1-4] are fundamentally different types of method, thus, directly comparing with the proposed method may not be a good idea. However, since they share the motivation and goal with this paper, I believe at least comparing with them in terms of the learned distance metric and/or the quality of sampled subgoal states might be helpful to guage the usefulness of the proposed method. Overall, the author's response (partly) addressed my main concerns and I'd like to increase my score from 5 to 6. ----- After author's response----- [1] Eysenbach, Ben, Russ R. Salakhutdinov, and Sergey Levine. "Search on the replay buffer: Bridging planning and reinforcement learning." Advances in Neural Information Processing Systems. 2019. [2] Huang, Zhiao, Fangchen Liu, and Hao Su. "Mapping state space using landmarks for universal goal reaching." Advances in Neural Information Processing Systems. 2019. [3] Zhang, Amy, et al. "Composable planning with attributes." International Conference on Machine Learning. 2018. [4] Savinov, Nikolay, Alexey Dosovitskiy, and Vladlen Koltun. "Semi-parametric topological memory for navigation." arXiv preprint arXiv:1803.00653 (2018). [5] Savinov, Nikolay, et al. "Episodic curiosity through reachability." arXiv preprint arXiv:1810.02274 (2018). Minor comments / questions In section 5.3, the difference between NegReward and HRAC is not clearly described. In Eq.(12), it seems HRAC also uses the adjacency constraint as a negative reward to the high-level policy.


Review 4

Summary and Contributions: Post-rebuttal: After reading the rebuttal and other reviews, I still strongly recommend acceptance. I think R3 did bring up some prior works that should probably have been cited, but I think this can be fixed by citing the missing works and more carefully stating the contribution of prior work. This paper tackles the problem of HRL by optimizing the high level policy to generate goals that are in a k-step adjacency region to the current state to reduce the effective space of high level goals. They prove that their method contains the optimal policy even with this k-step adjacency constraint, and the implement the method with a network to approximate the distance function between states, and a hinge loss function that optimizes the high-level reward to keep its subgoals within the k-step region.

Strengths: I think the core observations in the intro are really well said: HRL relies on good subgoals. The space of subgoals can end up being as large as the original state space, making HRL inefficient. You can reduce the action space, but you still need it to be expressive enough to form the optimal policy. Distant subgoals can be substituted by closer subgoals as long as they’re in the same direction. Figure 1 is a really good illustration of the main idea, which can seem pretty abstract otherwise. The experimental tasks are reasonably difficult actually, so I think the results are validated on actually difficult experimental settings. There is also a reasonably large number of different tasks. The method clearly outperforms reasonable HRL baselines. The adjacency visualizations in the appendix really show that this method is generating good subgoals and actually learning a distance function.

Weaknesses: I really don't have much to say here, it's really a good paper. I guess a question I have, not sure it's a weakness. Is using the old policy networks to approximate an expectation over all policies in equation 9 actually a good assumption? It appears to be in practice based on the results, but it seems like these policies are clearly not drawn iid. Any thoughts on this?

Correctness: I checked that Theorem 1 is true I checked that Theorem 2 is true The experimental methods seem sound

Clarity: The writing is incredibly clear and persuasive. I understood the intuition of the idea instantly from reading the introduction.

Relation to Prior Work: The related work section is quite thorough and also contrasts this method from prior work.

Reproducibility: Yes

Additional Feedback:

[Author Response · NeurIPS 2020]

We thank all reviewers for detailed and valuable comments, and will revise the paper accordingly as described below.

**Minor changes and typos.** We thank all reviewers for pointing those out, and will do corrections in the revision.

**R1 & R2: Discussions of deterministic MDPs in theoretical results.** Although our theoretical results are derived in
the context of deterministic MDPs, they are instructive for the practical algorithm design in general cases, and this
assumption is also exploited by some prior work that investigate distance metrics in MDPs (e.g. reference [11] in the
paper and the work of Castro mentioned by R1). Empirically, we have also showed that our method is robust to certain
kinds of stochasticity and outperforms the best baseline in stochastic environments (see line 40 – 48 and Figure 1).

**R1: Subgoals as short-term changes v.s. $k$-step adjacency constraint.** Interpreting the subgoals as short-term
changes amounts to setting a general constraint in the raw state space (e.g. HIRO) or in an embedding space (e.g.
FeUdal networks). However, the Euclidean distance between states in these spaces instead of the adjacency space may
not indicate the real adjacency relation: e.g. consider a grid-world environment where two states are separated by a wall.
Also, these general constraints control the maximum magnitude of the changes by a human-specified hyperparameter,
which is hard to choose in a principled way, while our method can learn the constraint automatically.

**R1: Claim of the empirical results.** We agree with the reviewer and will change the wording in the revision.

**R1: Comparison with two recent works.** (1) Castro proposes an algorithm for computing bisimulation metrics, which
reflect behavioral equivalence between states, using sampled transitions. Compared to our work, bisimulation metrics
depend on both the dynamics and the rewards, while the shortest transition distance depends only on the dynamics
and therefore can be easily applied to multi-task settings. (2) Khetarpal et al. present a theory and an algorithm of
affordances in RL, formulating the fact that certain states only enable certain actions. They construct the affordances
based on indents, i.e. desired state distributions, which are specified by humans a priori. In contrast, our method learns
subgoals, which can be interpreted as a kind of temporally extended indents, by RL rather than human prior.

**R2: Comparison with Options and EigenOptions with successor representation (SR).** (1) The Option framework
maintains a finite set of low-level Options (macro-actions), while our method (which falls into goal-conditioned HRL)
maintains an universal goal-reaching low-level policy whose behavior is modulated by subgoals. As shown in the
HIRO paper, goal-conditioned HRL often yields better performance than HRL with Options. (2) SR can be used to
measure the temporal distance between states and discover eigenoptions. Compared to our work, SR depends on both
environmental dynamics and a specific policy, while the shortest transition distance relies only on the dynamics.

**R2: Experimental settings.** We followed HIRO to use off-policy TD3 in continuous control tasks; in discrete control
tasks we found that whether using on-policy or off-policy methods (e.g. double DQN) does not make much differece.

**R3: Comparison with graph-based methods (missing baselines).** We believe that our method has essential difference
with graph-based methods: our method models high-level learning as a RL process and thus can tackle more general
problems, while current graph-based methods derive high-level policy *without* a training process, exploiting specific
problem structure. E.g. all graph-based works cited in the review obtain the subgoal sequence by solving a shortest-path
problem to a *known* goal node in a high-level graph (e.g. using Dijkstra's algorithm), which cannot be applied to
more general problems where there does not exist a single "goal" state (e.g. PointGather) or the goal state needs to be
explored by the agent instead of being given in advance (e.g. KeyChest), as in many of our experiments. To the best of
our knowledge, none of these graph-based works has reported results on these general problems without additional
task-specific prior. Therefore, we found it hard to fairly compare our method with graph-based methods in experiments
and thus did not add them to baselines. In the revision, we will add these discussions to the related work section.

**R3: Empirical study in stochastic MDPs.** To empirically verify the stochasticity robustness
of our method, we have applied HRAC to a set of stochastic AntMaze tasks, which have
relatively larger state (30-d continuous) and action space (8-d continuous) than the Maze task.
We added Gaussian noise with different STDs ($\sigma$) to the $(x, y)$ position of the ant robot at every
step. As shown in Figure 1, HRAC achieves similar asymptotic success rates with different
noise magnitudes. Due to the time limit, we only compare HRAC with the best baseline HIRO
when $\sigma = 0.1$, representing the noise magnitude that approximately equals to 20% of the
maximum step size of the ant robot on average, where HRAC achieves better performance than
HIRO. We will add more experimental results in stochastic environments to the revised paper.

Figure 1: Learning curves.

**R3: Difference between HRAC and NegReward.** HRAC plugs the adjacency loss as an extra
term into the loss of a specific RL algorithm, rather than treat it as a negative reward.

**R4: Explanation of the approximation in Eq. (9).** In Eq. (9), we apply a minimizing operation instead of an
expectation operation over a finite policy set to approximate the original minimizing operation over the whole (indefinite)
policy set. Therefore, we do not require that these policies are drawn i.i.d..

[Meta-Review · NeurIPS 2020]

All four reviewers recommended acceptance of this paper. Please incorporate the reviewer comments, particularly the missing references.